# Towards monitoring of global health research: an exploratory analysis of transparency and stakeholder engagement

Samruddhi Suresh Yerunkar [1,2] Nicole Hildebrand [2] Daniel Strech [1]

¹Berlin Institute of Health at Charité– Universitätsmedizin Berlin, QUEST Center for Responsible Research, Berlin, Germany
²Charité – Universitätsmedizin Berlin, Department of Psychiatry and Psychotherapy, Campus Benjamin Franklin, Berlin, Germany

**Correspondence to**
Dr Samruddhi Suresh Yerunkar;
samruddhi.yerunkar@charite.de

## ABSTRACT

**Objectives** Monitoring systems exist for clinical research transparency in high-income countries, but systematic assessment of these practices in global health (GH) research (GHR) is limited. We evaluated methods for monitoring GHR transparency and engagement.

**Design** Cross-sectional study.

**Data sources** Three sources were used: (a) ClinicalTrials. gov, (b) publications from 20 journals with 'international' or 'global' in the title and (c) outputs from selected GH funder websites.

**Eligibility criteria** From ClinicalTrials.gov, we selected 200 interventional trials on maternal health and tuberculosis (2008–2019), ensuring two-thirds were from low- and middle-income countries (LMIC). From journals, we included 200 trial publications (2011–2023). From funder websites, we included outputs with sufficient metadata to track trial registration and reporting.

**Data extraction and synthesis** Trials were extracted independently by two reviewers for result publications; journal articles were screened to confirm whether they reported trial results. Across all sources, we assessed registration timing, result reporting, open access and stakeholder engagement.

**Results** For 200 trials, 37% were prospectively registered, 65% published results in journals and 15% reported summary results in ClinicalTrials.gov. Only 34% reported results in any format within 24 months of completion. Of 200 publications, 72% were freely accessible, and 23% of the 100-article subsample included stakeholder engagement statements. The funder website sample yielded insufficient metadata for analysis.

**Conclusions** Monitoring GHR is feasible using registries and journals, though funder websites provide limited tracking. While open-access rates are encouraging, timely reporting and engagement documentation remain weak. These results highlight opportunities for developing GHR-specific monitoring approaches through collaborative efforts among global stakeholders.

## INTRODUCTION

Global health (GH) research (GHR) is a collaborative, transnational research approach aimed at promoting health equity for all.[1] The field addresses major health challenges that affect large populations worldwide, often focusing on severe conditions with substantial impact on mortality and quality of life, particularly in resource-limited settings. Recent surges in GHR activities, especially spurred by the COVID-19 pandemic, have highlighted both the field's potential and its challenges in delivering timely evidence for decision-making.

Research transparency and stakeholder engagement are crucial elements across all fields of health research. In GHR, these practices take on additional dimensions due to the field's unique characteristics. For instance, many GH conditions affect substantial populations across different regions and cultural contexts, making rapid knowledge translation particularly important. Limited research infrastructure in many settings and historical power imbalances between researchers and communities further emphasise the need for transparent research practices and meaningful stakeholder engagement.

## STRENGTHS AND LIMITATIONS OF THIS STUDY

⇒ We explored data from three distinct sources (clinical trial registries, global health (GH) journals and GH funder websites), introducing a methodology suitable for assessing GH research (GHR) studies using transparency and engagement indicators.

⇒ Our findings provide empirical data while highlighting the need for expanded collaboration to refine indicators and methodologies, enhancing the contextual relevance of GHR monitoring.

⇒ Trials from ClinicalTrials.gov were classified by location (low- and middle-income countries (LMIC) vs high-income countries), with two-thirds of selected trials from LMICs, ensuring equitable representation and addressing historical disparities in research quality assessments.

⇒ We have provided detailed steps, accompanying code and data to support reproducibility.

⇒ The study uses a sample of studies to demonstrate the feasibility of the methods, which can limit the generalisability of the findings.

The timely dissemination of research findings through transparent practices, such as prospective registration, comprehensive results reporting and open-access publication, is essential for evidence-based healthcare globally. In GHR, barriers to accessing research findings can be especially problematic—paywalled publications may be inaccessible to healthcare providers, researchers and patients in low-resource settings, potentially widening rather than reducing health inequities. Similarly, the transnational and cross-cultural nature of GHR makes genuine stakeholder engagement particularly relevant for ensuring research relevance, cultural appropriateness and eventual implementation.

Another defining characteristic of GHR is collaboration. The transnational and cross-sectoral nature of the field means that progress often depends on collective action across institutions, regions and disciplines. Such collaboration has already proven effective in addressing major health challenges. Examples include the scale-up of antiretroviral therapy and cervical cancer screening, which originated in high-income countries (HIC) and were later adapted to resource-limited settings.[2][3] In return, innovations pioneered and scaled from low- and middle-income countries (LMIC) under resource-constrained conditions—including the development of modern oral rehydration solution and the evolution of kangaroo mother care—have been adopted globally as low-cost, effective interventions.[4][5] Given this central role of collaboration, robust monitoring is essential to ensure accountability, transparency and meaningful engagement of stakeholders in order to maximise impact.

Historically, GH monitoring was primarily led by major intergovernmental agencies. However, academic research has increasingly complemented these efforts, enhancing the methods and rigour of GH measurement to support better decision-making by agencies, governments and donors.[6] While significant efforts have been made to monitor trial transparency and engagement in Europe and North America,[7–12] gaps remain in assessing these practices on a global scale.[13] For instance, the EU TrialsTrackers (https://eu.trialstracker.net/) specifically monitor whether summary results (SRs) are posted on the EU clinical trials register, and national-level dashboards have also been developed, providing a detailed breakdown of practices and helping stakeholders identify and address areas needing improvement.[14][15] Similar GHR monitors can help assess the current landscape, highlight actionable areas and guide stakeholders—including patients, researchers, physicians and global leaders—in making informed decisions.

Improved GH monitoring further requires new technologies and methods, as well as established norms and standards to facilitate global reporting.[16] To build an effective monitoring system for GH, it is important to first define what constitutes GH studies and assess them with appropriate indicators. We explored three methods for generating a sample of GH studies: (a) clinical trial registries focusing on GH conditions or diseases affecting populations globally, (b) publications from GH journals and (c) research outputs mentioned on GH funders' websites. Using indicators established in previous studies,[9–12] we assessed a sample of these studies to provide empirical data. We aim to demonstrate how past methodologies for monitoring clinical research can be applied to GHR, highlighting opportunities and challenges to enhance the implementation of monitoring systems in the GHR.

## METHODS

### Sampling strategy

We explored three complementary approaches to identify GHR studies: (a) clinical trial registries, using selected GH conditions as a proxy, (b) publications from journals with an explicit GH focus and (c) research outputs listed on GH funder websites. Each source offered distinct advantages and challenges for identifying and tracking GH studies. Below, we describe the methodology and purpose.

### Clinical trial registry with a disease-based approach

We systematically searched the trial registry of ClinicalTrials.gov for interventional trials with a start date from January 1, 2008 to a completion date of 31 March, 2019. This eligibility requirement was based on the mandate for clinical trial registration in registries under both US laws and journal policies, following International Committee of Medical Journal Editors (ICMJE) guidelines[17] passed on January 1, 2008. For our analysis, we focused on two health conditions known to have significant global burdens: maternal health issues (including postpartum depression, maternal sepsis and maternal anaemia) and tuberculosis. Trials were identified using relevant Medical Subject Headings (MeSH terms) for each condition (see online supplemental S1 for details). These conditions were chosen based on their significant GH burden and relevance to our ongoing larger research project, where one of our team members has a specific interest in maternal health, particularly postpartum conditions. We excluded international multicentric trials, observational studies and those with missing location or site information, as well as withdrawn studies. Trial locations were categorised as HIC or LMIC according to the World Bank country classification.[18] For our exploratory analysis, we randomly selected 100 trials per condition. To address the imbalance with more than half of the trials originating from HICs, we ensured that two-thirds of the selected trials were from LMICs. This approach ensured equitable representation and also addressed the historical disparity in research quality assessments between HICs and LMICs. In this sample, we aim to demonstrate the feasibility of evaluating registration practices and the timeliness of result reporting by examining both SRs and publications from a sample of clinical trials within a GH context.

To locate trial publications, two independent reviewers (NH and SSY) followed a predefined search manual

to identify the earliest publications on trial results (see online supplemental S2 for search manual). Searches were conducted in both the registry and Google, using trial IDs and relevant terms. Eligible publications included peer-reviewed articles or preprints with over 500 words, matched to the trial by design, population, intervention and comparator (if applicable). Additionally, the publication's primary outcome measure (or first-mentioned outcome) had to be listed as an outcome measure in the registration, whether classified as primary or not.

### GH journals

Our second approach focused on generating a sample of GH studies from peer-reviewed journals specialising in GH. We identified 20 journals that either had 'global health' or 'international health' in their title (see online supplemental S3 for the GH journals list). Using the Cochrane Highly Sensitive Search Strategy (Box 3b strategy), we located randomised trials published after January 1, 2011 in PubMed and cross-referenced these results with our selected journals. Including trial publications after 2011 allowed a 3-year window for dissemination of results, following the ICMJE mandate for prospective trial registration, effective from 2008. Our search yielded 1125 records of trials published after 2011. An author (SSY) screened a random sample (n=300), categorising them as primary reports of trial results or secondary analyses by the original or different trial investigators. For analysis, we included the first 200 primary reports and secondary analyses by the same investigators (see online supplemental S1 flowchart). This sample was used to evaluate how trial results are reported in GHR, examining adherence inclusion of data sharing and community engagement statements.

### GH funder website

The third method aimed to explore the use of GH funder websites as potential sources for generating GH studies. We investigated funder websites to determine if they provided sufficient metadata, such as trial registration numbers and registry links, to allow the tracking of funded clinical trials and to assess transparency. This involved searching World RePORT,[19] a database for tracking funding data for some of the largest biomedical funders, as well as a list of GH funders compiled by colleagues for a prior project. Our search revealed 20 GH funders, and 1 of them had explicit GH programmes with available funding metadata sufficient to track funded clinical trials (see online supplemental S4 for the GH funders list).

We focused on the Fogarty International Center at the National Institutes of Health (NIH), which supports GHR. Using their NIH RePORTER tool, we identified a list of 73 clinical trials and 9515 publications supported by 331 core projects. However, after applying the inclusion criterion with regard to the date, the number of relevant trials was very limited, leading us to exclude this sample from our analysis (see table 1 for an overview).

### Software

We used the aactr R package (V.0.0.0.9000[20]) to assess registration and SR reporting in the registry of trials. The R package ODDPub (V.7.2.2)[21] and roadoi (V.0.7.3[22]) were used to assess availability statements and open-access status, respectively, in trial publications. The map was created using the free and open-source Quantum Geographic Information System (QGIS) (V.7.2.2).[23] Data cleaning steps and statistical analyses were performed using R (V.4.3.2).[24] All analysis scripts are available under an open license on GitHub (https://github.com/quest-bih/prodigy), and the underlying data are accessible via the Open Science Framework at https://doi.org/10.17605/OSF.IO/HQDNS.[25]

### Patient and public involvement

Patients and/or the public were not involved in the design, conduct, reporting or dissemination plans of this research.

## RESULTS

This section provides an independent analysis of two samples: the first focuses on interventional trials from ClinicalTrials.gov, while the second examines trial results publications from GH journals. Data from the third funder website sample was limited and not analysed.

### Follow-up of clinical trials from clinical trial registry

We included data from 200 interventional trials (2009–2019) with 2/3rd representation from LMIC, enrolling 115 929 participants. Dominant trial characteristics were small sample size (<100) and open labelled (table 2). The most common trial locations were the USA (n=25), South Africa (n=24) and China (n=23) (figure 1).

### Trial registration and result reporting

Of the 200 trials, 74 (37%) were registered prospectively. Regarding result reporting, of the 200 trials that were completed in 2019 or earlier and thus had 4 years or more to report results, 129 trials (65%) had results published in journals and 30 (15%) reported SRs in registries. The number of trials that reported any results (either via journals or SRs) within 1, 2 or 4 years after completion was 29 trials (15%), 67 trials (34%) and 116 (58%) (table 3).

For journal result publications identified using a search manual, the reviewer independently located relevant publications, with an interrater reliability of 83%. When different publications were selected, 100% agreement was reached on selecting the earliest matching publication. All matched publications were peer-reviewed articles, with majority specifically, 60% (78/129), linked in the registry.

### Analysis of trial results in publications from GH journals

Of the 200 trial publications in GH journals from 2011 to 2023, 72% (144) were Gold Open Access (freely accessible), 22.5% (45) were Hybrid Open Access (partially accessible) and 1.5% (3) were closed access. We were

**Table 1**  Established indicators used for assessing transparency and stakeholder engagement in GHR samples

| Indicator | Details | Assessment | Impact |
|---|---|---|---|
| **Follow-up of clinical trials from the clinical trial registry with a disease-based approach** | | | |
| Time to registration | Before any clinical trial is initiated its details must be registered in a publicly available, free-to-access, searchable clinical trial registry complying with WHO's internationally agreed standards*. | Automated check by comparing the trial registration date with the trial start date. We report on<br>1. Before the trial starts (prospective registration).<br>2. Within 60 days after the trial starts (early registration).<br>3. After 60 days (late registration). | Transparency, unbiased reporting |
| Result reporting | Results can be reported in the following way: published articles in a peer-reviewed journal (24 months from trial completion); trial results uploaded in the trial registry (12 months from trial completion)*. | Automated check for SR reporting in the registry and manual search† for published articles. We report on<br>1. SR in registry.<br>2. Journal publication.<br>3. SR or journal publication within 1-year after trial completion.<br>4. SR or journal publication within 2 years after trial completion.<br>5. SR or journal publication within 4 years after trial completion. | Transparency, informed decision making |
| Open access | Publications describing clinical trial results should be open-access*. | Automated check for publication status (open access) for manually searched research publications of clinical trials. | Transparency, accessibility |
| **Analysis of clinical trial result publication in GH journals** | | | |
| Stakeholder/ community engagement | Involvement of patients and other relevant stakeholders (such as caregivers, healthcare providers, patient advocacy groups, policymakers and the community) in any phase of the clinical study process. | Manual review‡ of publications for stakeholder involvement statements. | Practical relevance, inclusivity |
| Open access | Publications describing clinical trial results should be open access*. | Automated check for publication status (open access). | Transparency, accessibility |
| Data sharing statement | Data sharing, archiving, deposition and accessibility of data in journal articles, with or without request. | Automated check for declarations on data sharing and availability. | Transparency, reproducibility |
| Code sharing statement | Support sharing of code whenever appropriate. | Automated check for open code statements. | Transparency, reproducibility |

*Text referenced from the WHO Joint Statement on Public Disclosure of Results from Clinical Trials (https://www.who.int/news/item/18-05-2017-joint-statement-on-registration).
†Manual searches for locating result publications are done by two according to the manual (see online supplemental S2).
‡Manual searches for screening studies with stakeholder engagement are done according to the manual (see online supplemental S5).
GH, global health; GHR, global health research; SR, summary result.

able to download 127 of the 200 PDFs to assess data and code availability. Among these, 59% (75) included a data-sharing statement, often specifying availability on request, and only 0.8% (1) included a code-sharing statement. A random sample of 100 articles was reviewed for stakeholder engagement statements using a predefined manual (see online supplemental S5 search manual), with 23% (23) reporting relevant content (table 4).

## DISCUSSION

This study aims to explore the approaches for monitoring transparency and stakeholder engagement in GHR while providing empirical data on the current state of these practices. We tested three different sampling strategies and found that trial registries focusing on GH conditions and GH journals provide viable data sources for monitoring, while funder websites currently lack sufficient metadata for systematic tracking.

Among 200 registered trials identified via ClinicalTrials.gov focusing on selected GH conditions, 37% were prospectively registered and 65% had published results

**Table 2** Characteristics of included clinical trials

| Trial characteristic* | Tuberculosis (n=100) (%) | Maternal health conditions (n=100) (%) | All trials, n=200 (%) |
|---|---|---|---|
| Recruitment status | | | |
| Completed | 79 (79) | 76 (76) | 155 (77.5) |
| Terminated | 2 (2) | 8 (8) | 10 (5) |
| Unknown status | 19 (19) | 16 (16) | 35 (17.5) |
| Enrolment | | | |
| 100 | 49 (49) | 40 (40) | 89 (44.5) |
| 100–500 | 32 (32) | 44 (44) | 76 (38) |
| >500 | 19 (19) | 16 (16) | 35 (17.5) |
| Masking (missing, n=1) | | | |
| Open label | 62 (62) | 53 (53) | 115 (57.5) |
| Single | 3 (3) | 16 (16) | 19 (9.5) |
| Double | 8 (8) | 9 (9) | 17 (8.5) |
| Triple | 7 (7) | 5 (5) | 12 (6) |
| Quadruple | 19 (19) | 17 (17) | 36 (18) |
| Design (missing, n=29) | | | |
| Non-randomised | 10 (10) | 15 (15) | 25 (12.5) |
| Randomised | 76 (76) | 70 (70) | 146 (73) |
| Phase | | | |
| I | 19 (19) | 5 (5) | 24 (12) |
| I–II | 3 (3) | 1 (1) | 4 (2) |
| II | 21 (21) | 14 (14) | 35 (17.5) |
| II–III | 2 (2) | 2 (2) | 4 (2) |
| III | 9 (9) | 13 (13) | 22 (11) |
| IV | 11 (11) | 24 (24) | 35 (17.5) |
| Not applicable | 35 (35) | 41 (41) | 76 (38) |

*Total counts may differ between categories because of missing data (number missing shown if >0).

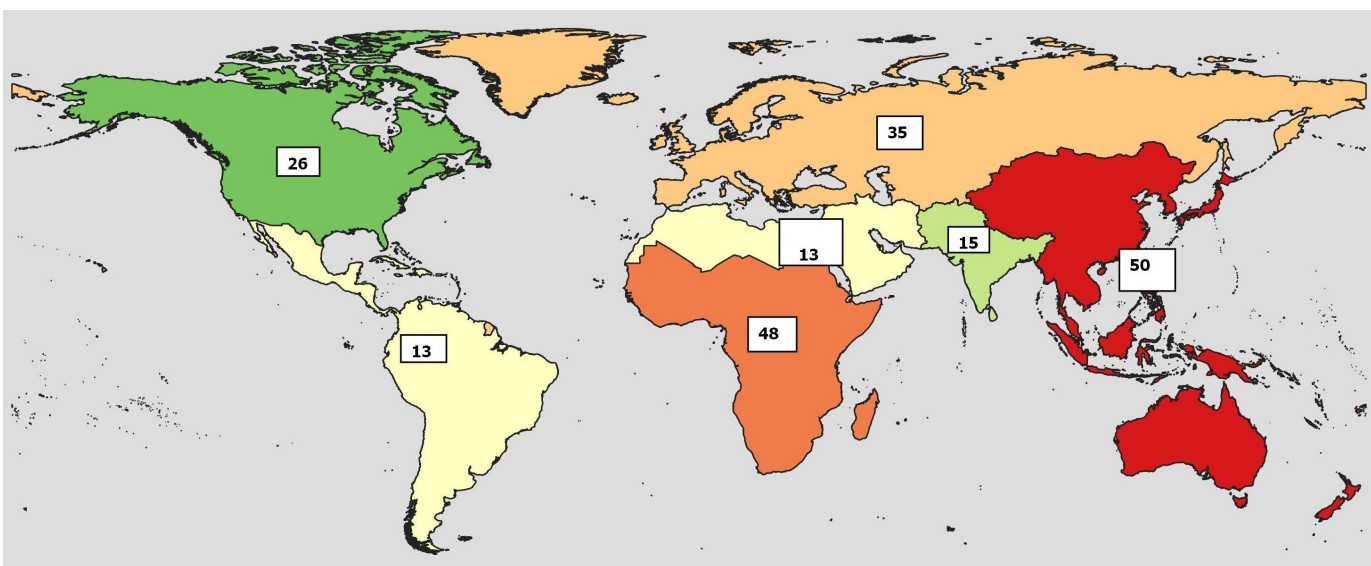

**Figure 1** Geographic distribution of included clinical trials.

**Table 3** Registration and result reporting of included clinical trials

| Indicators | Result (n, %) |
|---|---|
| | All trials, n=200 |
| Time to registration | |
| Prospective registration | 74 (37) |
| Within 60 days after trial start | 50 (25) |
| >60 days after trial start | 76 (38) |
| Result reporting | |
| Journal publication | 129 (65) |
| SR in registry | 30 (15) |
| Result reporting within (SR or journal) 1 year after completion | 29 (14.5) |
| Results reporting (SR or journal) within 2 years after completion | 67 (34) |
| Result reporting (SR or journal) within 4 years after completion | 116 (58) |
| Open access (n=125*) †Gold:22, green: 20, hybrid: 13, bronze:10, closed: 15, NA: 45 | 65 (52) |

*For all 129 journal publications, we found a DOI for 125 articles.
†Gold: articles published in fully open-access journals; green: articles self-archived in repositories or institutional archives; hybrid: articles published in subscription journals with an open-access option; closed: articles not publicly accessible.
SR, summary result.

in journals, though only 15% reported SRs. Within 24 months of trial completion, only 34% reported their results in any format and thus demonstrated timely results reporting. These findings on timely results reporting are comparable to previous studies examining clinical trial transparency in other contexts, such as German university medical centres (43%),[10] Nordic countries (52%)[9] and US academic medical centres (36%).[7] Among identified publications, 60% were linked to their registry

**Table 4** Transparency and community engagement in included trial result publication

| Indicators | Result (n, %) |
|---|---|
| Open access (n=200) *Gold: 144, green: 8, hybrid:45, closed:3. | 197 (99) |
| Data sharing statement, including 'available on request' (n=127†) | 75 (59) |
| Code sharing statement (n=127†) | 1 (0.8) |
| Stakeholder engagement statement (n=100‡) | 23 (23) |

*Gold: articles published in fully open-access journals; green: articles self-archived in repositories or institutional archives; hybrid: articles published in subscription journals with an open-access option; closed: articles not publicly accessible.
†For all 200, we had 127 PDFs.
‡Checked for a random sample of 100 publications.

entries, indicating moderate but improvable compliance with WHO recommendations for result traceability.[26] While the overall rates are similar to other contexts, timely results reporting is particularly important in GHR, given that many GH conditions represent a high global burden of disease. Although it would be problematic to claim that timely reporting is more important in some research fields than others, the often substantial disease burden addressed in GHR emphasises the need for rapid knowledge translation to inform evidence-based healthcare decisions.

For the sample of trial results publications from selected GH journals, the observed high rate of open-access publications (99%) is particularly encouraging, as paywalled research results would disproportionately affect healthcare providers and researchers in low-resource settings. This rate markedly exceeds the average of 28% open access found across scholarly literature.[27] While this high rate may partly reflect our sampling of GH journals with established open-access policies and increasing pressure on GH researchers to publish transparently, it nevertheless demonstrates substantial progress in making GHR findings widely accessible. The increasing adoption of institutional monitoring tools, as demonstrated by Franzen et al,[14] could help track and improve transparency in GHR. Such tools should incorporate core open science practices, recently identified through expert consensus, while considering GHR-specific needs.[28]

The baseline measurement of stakeholder engagement reporting from the GH journal sample—23% of the 100 publications assessed—demonstrates that evaluating GHR using this indicator is feasible and also provides important insights for future development. Given that GHR often involves transnational and cross-cultural research contexts, meaningful engagement of local communities and stakeholders is particularly crucial. Encouragingly, recent years have seen a positive push towards advancing patient and stakeholder engagement (PSE) in GHR. For example, initiatives such as the RESPIRE (Global Health Research Unit on Respiratory Health) group have developed structured protocols and practical tools, including templates for designing, implementing and evaluating stakeholder engagement plans that are tailored to local contexts and linked to planned outcomes. Such resources help translate field experiences into actionable guidance.[29] While our analysis establishes a baseline for Stakeholder engagement (PSE) reporting in GHR, more detailed analyses, such as those demonstrated by Weschke et al,[30] are needed to assess the quality and impact of these practices, providing valuable insights that can inform future improvements.

This study had several strengths and limitations. First, the disease-based sampling strategy using trial registries proved effective for identifying GHR studies but required careful selection of MeSH terms representing the conditions. A key methodological challenge was ensuring balanced representation between high- and low-income countries. Our approach of oversampling LMIC-based

studies (2/3 of the trial registry sample) provides one potential solution, though optimal sampling strategies for GHR monitoring should be further discussed with stakeholders. Second, the journal-based approach successfully identified GHR publications but is limited by the need to rely on 'global health journals' to represent GHR. For screening of open data and open code, we were able to automatically download 127 of 200 PDFs. These were then screened using the ODDPub tool, which has a reported sensitivity of 73%.[21] Manual validation was not performed—neither for downloading the remaining PDFs nor for identifying potential false negatives. This decision was consistent with our aim to demonstrate the feasibility of these methods in GHR monitoring in a scalable and often preferable automated manner, rather than to provide validated quantitative estimates. Finally, the funder-based approach highlighted gaps in tracking systems that limit comprehensive monitoring. While some GH funder websites provide lists of funded studies, they often lack essential metadata, such as trial registration numbers or the registry in which the trial is registered, which currently prevents automated assessment of transparency practices for these studies.

## CONCLUSIONS

In conclusion, our study revealed that monitoring GHR is feasible using trial registries and journals, although funder websites provide limited tracking. While open-access rates are encouraging, timely reporting and engagement documentation remain weak. We assessed only a sample of studies to provide empirical data on how GHR can be monitored, demonstrating the feasibility of these methods. However, GHR is inherently collaborative, requiring active input from researchers and stakeholders. Future efforts should identify additional indicators for monitoring, address challenges in informing researchers about their performance in transparency and expand on our findings using new samples and methodologies. Continued monitoring can drive improved practices, fostering more transparent and inclusive GHR to support evidence-based GH advancements.

**Acknowledgements** This study is part of the PRODIGY project, and we would like to thank Malek Bajbouj, Isabel Dziobek, Eric Hahn, Thi Minh Tam Ta and María Jose Lobeda-Garzón, as well as Sarah Weschke, Vladislav Nachev and Delwen Franzen, who contributed their ideas and knowledge to make this study possible.

**Contributors** SSY: conceptualisation, investigation, methodology, formal analysis, writing (original draft) and writing (review and editing). NH: investigation and writing (review and editing). DS: conceptualisation, methodology, writing (review and editing), supervision and funding acquisition. Guarantor for the overall content: SSY.

**Funding** This work was funded by the Berlin University Alliance (BUA), under grant number '812_Signature_Call_Prodigy'. DS received funding for a work package within the PRODIGY project. The funder had no role in the study design, data collection and analysis, decision to publish or preparation of the manuscript.

**Map disclaimer** The depiction of boundaries on this map does not imply the expression of any opinion whatsoever on the part of BMJ (or any member of its group) concerning the legal status of any country, territory, jurisdiction or area or of its authorities. This map is provided without any warranty of any kind, either express or implied.

**Competing interests** None declared.

**Patient and public involvement** Patients and/or the public were not involved in the design, conduct, reporting or dissemination plans of this research.

**Patient consent for publication** Not applicable.

**Ethics approval** Not applicable.

**Provenance and peer review** Not commissioned; externally peer reviewed.

**Data availability statement** Data are available in a public, open access repository. All the code for this study is available via open-source licensing on GitHub (https://github.com/quest-bih/prodigy). Data generated in this study are openly available under the Creative Commons Attribution 4.0 International license in OSF (https://doi.org/10.17605/OSF.IO/HQDNS).

**ORCID iDs**
Samruddhi Suresh Yerunkar https://orcid.org/0000-0002-0748-383X
Nicole Hildebrand https://orcid.org/0009-0003-5868-2614
Daniel Strech https://orcid.org/0000-0002-9153-079X

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
