## [Reviewer comments · BMJ Open]

ARTICLE DETAILS

Title (Provisional)

Towards Monitoring of Global Health Research: An Exploratory Analysis of Transparency and Stakeholder Engagement

Authors

Yerunkar, Samruddhi Suresh; Hildebrand, Nicole; Strech, Daniel

VERSION 1 - REVIEW

Reviewer	1
Name	Dinis, Maria Alzira Pimenta
Affiliation	Universidade Fernando Pessoa
Date	17-Apr-2025
COI	None

The manuscript is written in a scattered way, when all sentences need to be interlinked.

This style conflicts with references: “: (1) clin”

There is similarity to be addressed, then because preprints are all over and overlaps similarity.

This ended long ago, we are in 2025, so, the relevance is in question.

“2.1.1 Clinical Trial Registry with a disease-based approach: We systematically searched trial registry of ClinicalTrials.gov for interventional trials with start date between January 1, 2008 to completion date March 31, 2019.”

There are, in my opinion. Many formal aspects to be corrected, italics in variables, a scattered way to present the text (listed, not interconnected).

Discussion would need merge with results, it relies on a scarcity of references.

No conclusions section, clearly stating the context, results, findings, implications, limitations, prospects...

Reviewer	2
Name	Lahariya, Chandrakant
Affiliation	G. R. Medical College, Community Medicine
Date	07-Jun-2025
COI	None

1. Include strengths and limitations after discussion of the study.
2. Give some instances in introduction, where GHR has helped make major strides in public health. Can consider infectious/non-infectious diseases.
3. Mention the method separately used to filter out trials if any that were published in predatory journals.
4. Under software, provide detailed developer and version details as well.
5. In table 2, all forms of masking can be represented.
6. The break under section 3.2 about access to articles does not add up to 200 articles, necessary correction to be done.
7. Mention the strengths and limitations after or at the end of discussion.
8. Consider following references:
 - a. Mondal H, Mondal S. The paradigm shift in scientific publications. *Prev Med Res Rev* 2024;1:64-6. DOI: 10.4103/PMRR.PMRR_38_23
 - b. Saxena S, Tisdell E, Aumiller B. Perspectives of female physicians who lead in the public health sector in India – A qualitative study. *Prev Med Res Rev* 2024;1:241-7.
 - c. Rao KS. Public health and the role of the private sector. *Prev Med Res Rev* 2024;1:8-9. DOI: 10.4103/PMRR.PMRR_45_23

Reviewer	3
Name	Colby, Vorland
Affiliation	Indiana University Bloomington School of Public Health
Date	02-Sep-2025
COI	None

Yerunkar et al. evaluate methods for monitoring transparency and stakeholder engagement in global health research by analyzing studies from Clinicaltrials.gov and journal publications (and attempted to do so from health funder websites). They report that while open access

rates for publications are high, there are significant deficiencies in prospective trial registration and timely results reporting, which is perhaps (unfortunately) expected for non pharma trials. The article is well written, and though others may have made different decisions about how to design a study to answer these questions, I do not observe major flaws in the design or conclusions. Credit to the authors for sharing data and code. Below are minor comments.

The authors use an R package with various tools to automatically screen CT.gov and articles for some of the outcomes. I am familiar with some of these tools, but it would be of benefit to readers to have a brief description of how they were previously validated and what their performance was (for example, was validation done on PDFs, which introduces the potential for text extraction errors, for data and code statements).

2.1.1: What search strings were used specifically to search for those conditions on ClinicalTrials.gov? How many individuals screened and classified each result?

Table 2: it is not stated that the numbers in parentheses are percentages.

3.2: "We were able to download 127 of the 200 PDFs to assess data and code availability."

- Why could you not obtain the remainder through institutional access for a more representative sample? It could be that OA articles tend to have different data and code sharing statements.

Table 4 (note there are two table 2's - the latter needs to be renamed table 4): define gold, green, and hybrid open access minimally in the table legend.

VERSION 1 - AUTHOR RESPONSE

Reviewer 1

Responses to the comments made by Dr.Dinis,

1. Comment: "The manuscript is written in a scattered way, when all sentences need to be interlinked."

Response: We acknowledge that some sections of the manuscript may have appeared fragmented. To address this, we have revised the manuscript to improve the flow and coherence of the introduction and discussion section, as also noted in other comments. Reference styling and numbering have been standardized to ensure consistency with in-text citations. Additionally, URLs to the OSF repository have been replaced with direct citations to downloadable files, now included in the supplementary information. We hope that these revisions enhance the overall clarity and interconnectedness of the manuscript.

2. Comment: “This style conflicts with references: “: (1) clin”

Response: We have revised the conflicting reference numbering and updated it to use letters (a, b, c) for consistency with the reference style.

3. Comment: There is similarity to be addressed, then because preprints are all over and overlaps similarity.

Response: Thank you for pointing out these issues. We have updated the references as follows:

Reference 5: The preprint has been replaced with the recently published article.

Reference 12: The faulty URL has been corrected.

All references: Standardized according to the recommended software including version number/style guidelines.”

4. Comment: This ended long ago, we are in 2025, so, the relevance is in question. 2.1.1 Clinical Trial Registry with a disease-based approach: We systematically searched trial registry of ClinicalTrials.gov for interventional trials with start date between January 1, 2008 to completion date March 31, 2019.”

Response: Thank you for this feedback. This was a deliberate choice, as we wanted to allow at least four years or more for trials to publish their results by the time of our analysis in 2024. We have highlighted this point again in the manuscript for readers’ clarity. Additionally, the focus of this study was on exploring methodologies to assess global health research using different indicators, rather than providing the statistics themselves. We hope that future studies could build on this work by applying these methods in the context of global health research to generate updated statistics.

5. Comment: There are, in my opinion. Many formal aspects to be corrected, italics in variables, a scattered way to present the text (listed, not interconnected).

Response: Thank you for this critical feedback. We appreciate it and have carefully revised the manuscript to address these formal issues, including standardizing italics for variables and improving the flow of the text.

6. Comment: “Discussion would need merge with results, it relies on a scarcity of references.No conclusions section, clearly stating the context, results, findings, implications, limitations, prospects...”

Response: Point noted. We have revised the discussion section to summarize the findings and their implications, incorporating additional relevant references to support key points. Strengths and limitations have been moved to follow the main discussion, and a conclusion section has been added at the end to clearly state the context, findings, implications, and future prospects.

Reviewer 2

Responses to the comments made by Dr. Lahariya,

1. Comment: Include strengths and limitations after discussion of the study.

Response: Point noted. We have revised the discussion section as follows: summarising findings and their implications with additional references to the literature, moving strengths and limitations to follow the main discussion, and adding a conclusion section at the end.

2. Comment: Give some instances in introduction, where GHR has helped make major strides in public health. Can consider infectious/non-infectious diseases.

Response: Thank you for this valuable suggestion. We acknowledge that highlighting successful instances of GHR can strengthen the introduction by showing the impact of global health collaboration and innovation. We have therefore added the following lines to the introduction: “Another defining characteristic of GHR is collaboration. The transnational and cross-sectoral nature of the field means that progress often depends on collective action across institutions, regions, and disciplines. Such collaboration has already proven effective in addressing major health challenges. Examples include the scale-up of antiretroviral therapy and cervical cancer screening, which originated in high-income countries (HIC) and were later adapted to resource-limited settings (2,

3). In return, innovations pioneered and scaled from LMICs under resource-constrained conditions - including development of modern oral rehydration solution, evolution of kangaroo mother care - have been adopted globally as low-cost, effective interventions (4, 5). Given this central role of collaboration, robust monitoring is essential to ensure accountability, transparency, and meaningful engagement of stakeholders in order to maximize impact.”

3. Comment: Mention the method separately used to filter out trials if any that were published in predatory journals.

Response: We selected journals that have a clear focus on global health and are peer-reviewed. The journal list was manually validated. For this reason, no additional filtering for predatory journals was done. A table of the selected global health journals has been included in the supplemental file S3 for reference.

4. Comment: Under software, provide detailed developer and version details as well.

Response: Thank you for this comment. We have revised the software section in the references to include developer details, version numbers, references to R packages, and URLs to their GitHub pages. Version details have also been added to the in-text citations to ensure clarity.

5. Comment: In table 2, all forms of masking can be represented.

Response: This is a nice observation. We have now represented all forms of masking (open label, single, double, triple, quadruple, NA) in Table 2. Characteristics of included clinical trials.

6. Comment: The break under section 3.2 about access to articles does not add up to 200 articles, necessary correction to be done.

Response: There was an extra comma, which has now been removed. The breakdown under open access (144, 8, 45, 3) adds up to 200.

7. Comment: Mention the strengths and limitations after or at the end of discussion. Consider following references:

a. Mondal H, Mondal S. The paradigm shift in scientific publications. *Prev Med Res Rev* 2024;1:64-6. DOI: 10.4103/PMRR.PMRR_38_23

b. Saxena S, Tisdell E, Aumiller B. Perspectives of female physicians who lead in the public health sector in India – A qualitative study. *Prev Med Res Rev* 2024;1:241-7.

c. Rao KS. Public health and the role of the private sector. *Prev Med Res Rev* 2024;1:8-9. DOI: 10.4103/PMRR.PMRR_45_23:

Response: Thank you for this comment. As noted in response to another comment, we have revised the discussion section to summarize the findings and their implications, incorporating additional relevant references. Strengths and limitations have been moved to follow the main discussion, and a conclusion section has been added at the end.

Response to Reviewer 3

Responses to the comments made by Prof. Colby,

1. Comment: “Yerunkar et al. evaluate methods for monitoring transparency and stakeholder engagement in global health research by analyzing studies from Clinicaltrials.gov and journal publications (and attempted to do so from health funder websites). They report that while open access rates for publications are high, there are significant deficiencies in prospective trial registration and timely results reporting, which is perhaps (unfortunately) expected for non pharma trials. The article is well written, and though others may have made different decisions about how to design a study to answer these questions, I do not observe major flaws in the design or conclusions. Credit to the authors for sharing data and code. Below are minor comments”.

Response: Thank you for your kind words!

2. Comment: “The authors use an R package with various tools to automatically

screen CT.gov and articles for some of the outcomes. I am familiar with some of these tools, but it would be of benefit to readers to have a brief description of how they were previously validated and what their performance was (for example, was validation done on PDFs, which introduces the potential for text extraction errors, for data and code statements)”

Response: Thank you for bringing this to our attention. Following this comment, we have revised the discussion section by adding these lines: “For screening of open data and open code, we were able to automatically download 127 out of 200 PDFs. These were then screened using the ODDPub tool, which has a reported sensitivity of 73% (18). Manual validation was not performed - neither for downloading the remaining PDFs nor for identifying potential false negatives. This decision was consistent with our aim to demonstrate the feasibility of these methods in GHR monitoring in a scalable and often preferable automated manner, rather than to provide validated quantitative estimates”.

3. Comment: “2.1.1: What search strings were used specifically to search for those conditions on ClinicalTrials.gov? How many individuals screened and classified each result?”

Response: Response: Thank you for this observation. The search strings (MeSH terms) used to identify the conditions on ClinicalTrials.gov were added to the supplementary file ‘Supplement S1: Search strategies to identify global health studies. In Section 2.1.1, we also now explicitly mention that these conditions were searched using the specified search strings and cite the relevant supplementary table for reference (“Trials were identified using relevant MeSH terms for each condition”). Regarding the number of individuals who screened and classified each result, we have highlighted the following statement in the methods as well as abstract section: “To locate trial publications, two independent reviewers (NH and SSY) followed a predefined search manual to identify the earliest publications of trial results (see Supplement S2 for the search manual).”

4. Comment: “Table 2: it is not stated that the numbers in parentheses are percentages.”

Response: Thank you for pointing this out. For clarification, we have added the percentage symbol (%) to Table 2, Characteristics of included clinical trials.

5. Comment: “3.2: "We were able to download 127 of the 200 PDFs to assess data and code availability."- Why could you not obtain the remainder through institutional access for a more representative sample? It could be that OA articles tend to have different data and code sharing statements.”

Response: This is a valid point that was discussed during the study. We decided against it because our primary focus was on the feasibility of the methods rather than on providing exact quantitative results. To acknowledge this, we have now added the following line to our limitations to highlight this point.

“For screening of open data and open code, we were able to automatically download 127 out of 200 PDFs. These were then screened using the ODDPub tool, which has a reported sensitivity of 73% (18). Manual validation was not performed - neither for downloading the remaining PDFs nor for identifying potential false negatives. This decision was consistent with our aim to demonstrate the feasibility of these methods in GHR monitoring in a scalable and often preferable automated manner, rather than to provide validated quantitative estimates.

6. Comment: “Table 4 (note there are two table 2's - the latter needs to be renamed table 4): define gold, green, and hybrid open access minimally in the table legend.

Response: Thank you for spotting this. I have renumbered the tables accordingly and added a minimal legend defining the different open access statuses.

VERSION 2 - REVIEW

Reviewer	3
Name	Colby, Vorland
Affiliation	Indiana University Bloomington School of Public Health
Date	11-Nov-2025
COI	

Thank you for addressing my comments.